# Mechanism of dimer selectivity and binding cooperativity of BRAF inhibitors

Joseph Clayton[1,2], Aarion Romany[1], Evangelia Matenoglou[3], Evripidis Gavathiotis[3], Poulikos I Poulikakos[4], Jana Shen[1]*

[1]Department of Pharmaceutical Sciences, University of Maryland School of Pharmacy, Baltimore, United States; [2]Division of Applied Regulatory Science, Office of Clinical Pharmacology, Office of Translational Sciences, Center for Drug Evaluation and Research, United States Food and Drug Administration, Silver Spring, United States; [3]Department of Biochemistry, Department of Medicine, Department of Oncology, Montefiore Einstein Comprehensive Cancer Center, Albert Einstein College of Medicine, New York, United States; [4]Department of Oncological Sciences, Icahn School of Medicine at Mount Sinai, New York, United States

*For correspondence:
jana.shen@rx.umaryland.edu

Competing interest: The authors declare that no competing interests exist.

## eLife Assessment

This **fundamental** study illuminates the dynamics of BRAF in its monomeric and dimeric forms, both in the absence and presence of inhibitors, through a **convincing** combination of traditional experiments and sophisticated computational analyses. By revealing novel insights into the selectivity and cooperative processes of BRAF inhibitors, it holds significant promise for the development of future therapeutics, particularly against mutant isoforms in cancer. Overall, these findings will be of great interest to structural biologists, medicinal chemists, and pharmacologists.

**Abstract** Aberrant signaling of BRAF$^{V600E}$ is a major cancer driver. Current FDA-approved RAF inhibitors selectively inhibit the monomeric BRAF$^{V600E}$ and suffer from tumor resistance. Recently, dimer-selective and equipotent RAF inhibitors have been developed; however, the mechanism of dimer selectivity is poorly understood. Here, we report extensive molecular dynamics (MD) simulations of the monomeric and dimeric BRAF$^{V600E}$ in the apo form or in complex with one or two dimer-selective (PHI1) or equipotent (LY3009120) inhibitor(s). The simulations uncovered the unprecedented details of the remarkable allostery in BRAF$^{V600E}$ dimerization and inhibitor binding. Specifically, dimerization retrains and shifts the αC helix inward and increases the flexibility of the DFG motif; dimer compatibility is due to the promotion of the αC-in conformation, which is stabilized by a hydrogen bond formation between the inhibitor and the αC Glu501. A more stable hydrogen bond further restrains and shifts the αC helix inward, which incurs a larger entropic penalty that disfavors monomer binding. This mechanism led us to propose an empirical way based on the co-crystal structure to assess the dimer selectivity of a BRAF$^{V600E}$ inhibitor. Simulations also revealed that the positive cooperativity of PHI1 is due to its ability to preorganize the αC and DFG conformation in the opposite protomer, priming it for binding the second inhibitor. The atomically detailed view of the interplay between BRAF dimerization and inhibitor allostery as well as cooperativity has implications for understanding kinase signaling and contributes to the design of protomer selective RAF inhibitors.

**Figure 1.** The X-ray structure of the BRAF[V600E] dimer in complex with PHI1. Left. Cartoon representation of the BRAF[V600E] dimer in complex with PHI1 (PDB: 6P7G *Cotto-Rios et al., 2020*, two protomers are colored tan and grey). The αC-helix, a-loop, and c-loop are colored orange, yellow, and pink, respectively. Right. A zoomed-in view of a PHI1-bound protomer. PHI1 and the sidechains of DFG-Asp594, αC-Glu501, catalytic Lys483, and HRD-His574 are shown as sticks.

The online version of this article includes the following figure supplement(s) for figure 1:

**Figure supplement 1.** Visualization of the H510-H477-D595 interactions at the dimer interface of BRAF[V600E].

**Figure supplement 2.** Protonation and tautomer states of the interface histidines determined by the all-atom PME-CpHMD titration.

**Figure supplement 3.** Convergence of the protonation/tautomer states of H477 and H510 at pH 7.5.

## Introduction

The mitogen activated protein kinase (MAPK) signaling cascades regulate cell growth, proliferation, and survival in mammalian cells (*Samatar and Poulikakos, 2014*; *Lavoie et al., 2020*). In the well-studied Ras-Raf-MEK-ERK pathway, the GTP-loaded RAS contacts RAF and induces its dimerization; the newly formed RAF dimer phosphorylates MEK which in turn phosphorylates ERK, which then phosphorylates a number of downstream proteins and regulate their functions (*Lavoie et al., 2020*). Mutations of BRAF, a kinase within the RAF family, are present in about 8% of human tumors, most commonly melanoma and colorectal cancers, with the mutation V600E accounting for about 90% of them. It is believed that the wild type BRAF signals as a dimer, the BRAF[V600E] is able to signal as a monomer (*Poulikakos et al., 2010*; *Poulikakos et al., 2011*; *Karoulia et al., 2017*). The first generation BRAF[V600E] inhibitors, including the current FDA-approved inhibitors, Vermurafenib, Dabrafenib, and Encorafenib, inhibit the monomeric BRAF[V600E]; however, drug resistance led to only short-term cancer remission in patients (*Poulikakos et al., 2011*; *Peng et al., 2015*; *Monaco et al., 2021*; *Yen et al., 2021*; *Adamopoulos et al., 2021*).

In the adaptive drug resistance mechanism, RAF dimerization renders the monomer-selective inhibitors ineffective (*Poulikakos et al., 2011*; *Peng et al., 2015*; *Monaco et al., 2021*; *Yen et al., 2021*; *Adamopoulos et al., 2021*). To overcome the resistance, inhibitors that are either dimer selective or equipotent to both monomers and dimers have been developed and entered clinical development (*Adamopoulos et al., 2021*; *Cook and Cook, 2021*). Understanding the molecular mechanism of dimer selectivity would be valuable for the rational design of RAF inhibitors. Although several MD studies have examined the conformational dynamics of BRAF[V600E] monomer (*Maloney et al., 2021*), BRAF[V600E] in complex with the monomer-selective inhibitors (*Tse and Verkhivker, 2016*), and wild type RAF dimerization (*Zhang et al., 2021*), the topic of RAF dimer selectivity has not been explored.

The kinase domain of the BRAF monomer has a typical kinase structure: a primarily β-sheet N-terminal domain connected to a helical C-terminal domain by a flexible hinge (*Figure 1*). Like other

kinases, the catalytic activity of BRAF depends on the conformation of two motifs: the αC-helix, which contains the conserved residue Glu501, and the DFG motif on the activation loop (a-loop), which contains the conserved ATP-binding (via magnesium) residue Asp594. In the active state, both the αC helix and DFG adopt the IN conformation, dubbed CIDI. In this state, the αC helix is positioned inward such that αC-Glu501 and the catalytic Lys483 form a salt bridge; meanwhile the DFG motif is also IN, meaning DFG-Asp594 is near Lys483 often in a salt-bridge distance. An inactive conformation can be achieved if either or both the αC helix and DFG motif adopt an OUT state. Specifically, αC-out involves an outward movement of the αChelix, while DFG-out involves the sidechains of the DFG Asp594 and Phe595 exchanging regions, that is Phe595 facing the ATP binding site and Asp594 facing the αC-helix.

In the BRAF dimer, the two protomers are arranged side by side and the dimer interface involves the C-terminal end of the αC helix (*Figure 1*). Current monomer-selective BRAF$^{V600E}$ inhibitors bind in the αC-out conformation, whereas the dimer-selective or equipotent inhibitors bind in the αC-in conformation (*Supplementary file 1A*). Thus, the αC conformation has been the center of attention in numerous structural and biochemical studies to understand RAF signaling and inhibitor activities (*Rajakulendran et al., 2009*; *Thevakumaran et al., 2015*; *Karoulia et al., 2016*).

In a recent study, Gavathiotis and coworkers discovered a modification to the dimer-compatible inhibitor Ponatinib which can increase the dimer selectivity by more than three fold (*Cotto-Rios et al., 2020*). The novel inhibitor, named Ponatinib hybrid inhibitor 1 (PHI1), extends the headgroup of Ponatinib by replacing the methylpiperazine with the 4-(2-aminoethyl) morpholino group. Remarkably, PHI1 showed more potent inhibition of the second protomer in the BRAF$^{V600E}$ dimer; in contrast, Ponatinib and equipotent inhibitors, for example LY3009120 or LY, AZ-628, and TAK-632, are non-cooperative (*Cotto-Rios et al., 2020*). The co-crystal structure of BRAF$^{V600E}$ in complex with PHI1 (PDB: 6P7G) (*Cotto-Rios et al., 2020*) revealed that the morpholine group extends the ligand-kinase interaction from the type-II pocket (occupied by all DFG-out inhibitors) to the center of αC helix, allowing a hydrophobic interaction with Asn500 next to the αC-Glu501 (*Figure 1*). The co-crystal structures show that this interaction is not available with the shorter Ponatinib (PDB ID: 6P3D; *Cotto-Rios et al., 2020*) or equipotent inhibitors, for example LY3009120 (LY, PDB ID: 5C9C; *Peng et al., 2015*). Gavathiothis and coworkers noticed that PHI1 stabilizes the αC helix in a slightly different IN conformation as compared to Ponatinib and hypothesized that the additional interaction with Asn500 is a key to the dimer selectivity of PHI1, as it may be unfavorable in monomer binding (*Cotto-Rios et al., 2020*). Shortly after, a biochemical study supported by the molecular dynamics (MD) simulations suggested that restriction of the αC helix movement is the basis for the difference between dimer-selective and equipotent inhibitors (*Adamopoulos et al., 2021*); however, the detailed mechanism remains elusive.

Prompted by the open questions regarding dimer selectivity and binding cooperativity of BRAF$^{V600E}$ inhibitors, we carried out a series of all-atom molecular dynamics (MD) simulations to investigate the conformational dynamics of the monomeric and dimeric BRAF$^{V600E}$ in the presence and absence of one or two dimer-selective (PHI1) or equipotent (LY) inhibitor(s). Analysis of the simulation data which ammounts to 135 µs aggregate time uncovered the atomic details of the remarkable conformational allostery in BRAF$^{V600E}$ dimerization and inhibitor binding. Supported by the co-crystal structure analysis of the published monomer-selective, dimer-selective, and equipotent inhibitors, an atomically detailed mechanism emerged that explains the monomer or dimer selectivity and binding cooperativity of BRAF$^{V600E}$ inhibitors. The mechanism also led us to propose an empirical method based on the co-crystal structure for assessing the dimer selectivity of BRAF$^{V600E}$ inhibitors.

## Results and discussion
### Analysis of the co-crystal structures suggests the h-bond formation with αC-Glu501 as a key requirement for dimer binding

To understand the preference of BRAF$^{V600E}$ inhibitors for the monomer vs. dimer form, we first examined all published co-crystal structures in complex with the monomer-selective and dimer-compatible (i.e. dimer-selective and equipotent) inhibitors (see *Supplementary file 1, table 1* for a complete list). We first noticed that the monomer-selective inhibitors, for example Vemurafenib (VEM, PDB ID: 5JRQ; *Grasso et al., 2016*), do not occupy BP-III, whereas most dimer-compatible inhibitors do. This can be explained by the observation that the monomer-selective inhibitors bind

in the DFG-in, whereas most dimer-compatible inhibitors bind in the DFG-out conformation–BP-III is occupied by Phe595 in the DFG-in conformation, so the pocket is only available in the DFG-out conformation (*Figure 2a*). Note, the equipotent inhibitor SB590885 (PDB ID: 2FB8; *King et al., 2006*) does not occupy BP-II or BP-III, as it binds in the DFG-in conformation (*Supplementary file 1, table 1*).

The co-crystal structure analysis revealed an important distinction between the monomer-selective and dimer-compatible inhibitors, namely, the former binds in the αC-out whereas the latter binds in the αC-in conformation. The interaction fingerprints showed that while most monomer-selective inhibitors make a hydrophobic contact with Leu505 next to the conserved RKTR motif at the end of the αC helix, only the dimer-compatible inhibitors interact with αC-Glu501 by donating a h-bond (e.g. from an amide group in PHI1 and LY) to the carboxylate sidechain of Glu501 (*Figure 2b*). Glu501 rests above BP-II in the DFG-out conformation (called BP-II-out) and may interact with the catalytic Lys483 (see later discussion), which makes up a part of BP-I. Interestingly, even though SB590885 binds in the DFG-in conformation, it can also donate a h-bond to Glu501 through an oxime hydroxyl group (PDB ID: 2FB8; *King et al., 2006*). This h-bond stabilizes the salt-bridge between the catalytic Lys483 and Glu501 such that the αC helix position is further inward (according to the KLIFs definition, see later discussion) as compared to the co-crystal structures in complex with other dimer-compatible inhibitors (*Supplementary file 1A*).

All monomer-selective and dimer-compatible inhibitors interact with the DFG-Asp594 although with subtle differences.

In PHI1 (PDB ID: 6P7G) and LY (PDB ID: 5C9C), the amide carbonyl occupying the BP-II accepts a h-bond from the backbone amide of Asp594, while in VEM (PDB ID: 4RZV; *Grasso et al., 2016*) the sulfonamide occupying the BP-II donates a h-bond to the backbone amide of Asp594 (orange in *Figure 2a*). One unique property of PHI1 is the ability to donate a h-bond to the backbone carbonyl of HRD-His574 through an amino nitrogen next to the morpholine headgroup. This region is classified as BP-IV by KLIFS, although the sidechain of His574 is a part of BP-II (in PDB 6P7G) and makes a h-bond with the backbone of the DFG-1 Gly593. Among the other dimer-compatible inhibitors, only Ponatinib (PDB ID: 6P3D; *Cotto-Rios et al., 2020*; *Adamopoulos et al., 2021*) makes a similar h-bond with the backbone of His574 through the methyl pyrazine headgroup.

In addition to analyzing the co-crystal structures, we also tested the inhibition of ERK1/2 phosphorylation in two melanoma cell lines by PHI1, LY, or VEM (*Figure 2c and d*). SKEML239 expresses monomeric BRAF[V600E], while SKMEL239-C4 expresses dimeric BRAF[V600E] (*Cotto-Rios et al., 2020*). Among the three inhibitors tested, PHI1 is the only compound to be more potent against SKMEL239-C4 versus SKMEL239 (IC50 of 256 nM vs. 1.5 μM). By contrast, LY has similar potency (27 nM vs. 15 nM) while VEM is more potent against SKMEL239 (3 μM vs 35 nM). This data confirms that PHI1 is dimer-selective, LY equipotent, and VEM monomer-selective.

In light of the above finding and given the central location of Glu501 on the αC helix, we hypothesized that the ability to form a h-bond with Glu501 is required by dimer-compatible inhibitors, as the h-bonding would restrict the αC helix to the αC-in conformation as observed in the co-crystal structures of all dimer-compatible inhibitors. This restriction was also suggested as a key for dimer selectivity in the recent study by Poulikakos and coworkers (*Adamopoulos et al., 2021*). However, the crystal structures do not provide an explanation for why the ability to induce the αC-in conformation enables the inhibitor to favor dimeric BRAF[V600E] over monomeric BRAF[V600E]. Thus, to test the hypothesis regarding the role of h-bond with the Glu501 and to dissect the mechanism of dimer selectivity, we conducted a series of MD simulations of the monomeric and dimeric BRAF[V600E] in the absence and presence of two dimer-compatible inhibitors (see below).

## Overview of the MD simulations of the monomeric and dimeric BRAF[V600E]

The dimer interface of BRAF[V600E] contains two histidines, His477 and His510. His510 forms a h-bond with His477 of the opposite protomer, while His477 is also in a potential salt bridge distance from Asp595 of the opposite protomer (*Figure 1—figure supplement 1*).

In a preliminary simulation, where all histidines were set to be neutral and in a tautomer state determined by inspection of the X-ray structure (His477 was set to HID; all others set to the AMBER *Case et al., 2020* default HIE), we found the BRAF[V600E] dimer dissociated within a few hundred nanoseconds.

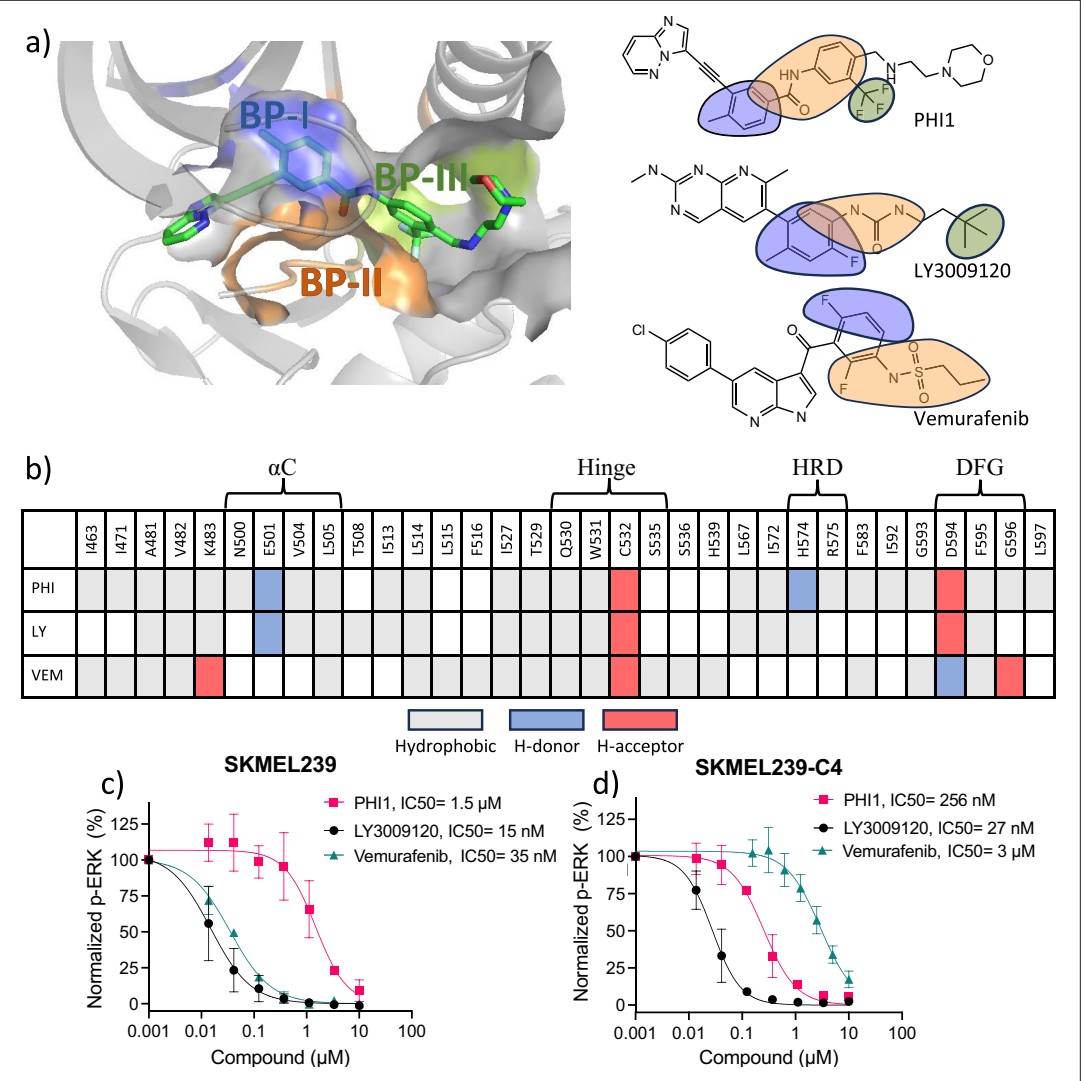

**Figure 2.** Protein-ligand interaction fingerprints for PHI1, LY3009120, Vermurafenib, and the inhibition of ERK1/2 phosphorylation in melanoma cells. a) Left. Visualization of the back pockets (BPs) in BRAF[V600E] in complex with PHI1. BP-I, BP-II, and BP-III are colored blue, orange, and green, respectively. BP definitions of *Liao, 2007*) are followed. (**a**) Right. Chemical structures of the example dimer selective (PHI1), equipotent (LY3009120 or LY), and monomer selective (Vermurafenib or VEM) inhibitors of BRAF[V600E]. Portions of structures are highlighted according to the BPs they occupy in the co-crystal structure (PDB IDs: 6P7G, 5C9C, and 4RZV). (**b**) Protein-ligand interaction fingerprints for PHI1, LY, and VEM in BRAF[V600E] according to the co-crystal structures (PDB IDs: 6P7G, 5C9C, and 4RZV). White indicates no interaction, while grey, blue, and red indicate hydrophobic, h-bond donor (H-donor) and acceptor (H-acceptor) interactions, respectively. These interactions were calculated by KLIFS (*Kooistra et al., 2016*) and manually verified and corrected. A h-bond was defined using the donor-accept distance cutoff of 3.5 Å, and a hydrophobic contact cutoff of 4 Å was used for aromatic interactions and 4.5 Å for non-aromatic interactions. For simplicity, aromatic face-to-face interactions are indicated as hydrophobic. An extensive list of monomer-selective and dimer-compatible inhibitors with co-crystal structures is given in *Supplementary file 1A*. (**c,d**) Inhibition of ERK1/2 T202/Y204 phosphorylation in SKMEL239 (**c**) and SKMEL239-C4 (**d**) melanoma cells (50,000 cells/well) following one hour treatment at 37° C by PHI1, LY3009120, and Vermurafenib in different concentrations. Normalized values and non-linear regression fits of ERK phosphorylation % are shown for different compounds. Error bars represent mean ± SEM with n=3.

The online version of this article includes the following figure supplement(s) for figure 2:

**Figure supplement 1.** Chemical structures of the monomer-selective, equipotent, and dimer-selective BRAF[V600E] inhibitors in *Supplementary file 1A*.

**Table 1.** Summary of the fixed-protonation-state MD simulations (aggregate time of 135 μs).

| No. | System | Simulation time | Starting structure |
|---|---|---|---|
| 1 | Apo monomer | 3x5μs | 6P7G(A) |
| 2 | Apo dimer | 3x5μs | 6P7G (inhibitors removed) |
| 3 | Holo monomer:PHI1 | 6x5μs | 6P7G (A:PHI1 or B:PHI1) |
| 4 | Holo monomer:LY | 3x5μs | 5C9C (A:LY) |
| 5 | Mixed dimer:PHI1 | 3x5μs | 6P7G (apo A; B:PHI1) |
| 6 | Holo dimer:2PHI1 | 3x5μs | 6P7G (A:PHI1, B:PHI1) |
| 7 | Mixed dimer:LY | 3x5μs | 5C9C (A:LY; apo B) |
| 8 | Holo dimer:2LY | 3x5μs | 5C9C (A:LY; B:LY) |

To rigorously determine protonation states, we applied the all-atom continuous constant pH molecular dynamics (CpHMD) titration (*Harris et al., 2022*), which revealed that His477 is most likely in the charged HIP state while His510 is most likely in the neutral HIE state at neutral pH (*Figure 1—figure supplements 2 and 3*).

Based on the CpHMD determined protonation states, we carried out a series of fixed-charge MD simulations of the monomeric and dimeric BRAF$^{V600E}$ in the ligand-free state (apo) or in complex with the PHI1 or LY inhibitor in each protomer (holo). To investigate the cooperativity of inhibitor binding, MD simulations were also conducted where only one protomer is complexed with the PHI1 or LY inhibitor (mixed). Each simulation lasted 5 μs and was repeated three times for statistical significance; in total, 135 μs trajectory data was collected (*Table 1*) and the last 3 μs of each repeat was used for analysis.

We note that enhanced sampling methods were not used due to several challenges. First, the BRAF dimer is weakly associated, with αC helix forming a part of the dimer interface (*Figure 1a*). Enhanced sampling (particularly of αC helix) would likely lead to dimer dissociation. Second, biased sampling methods such as metadynamics may lead to unrealistic conformational states due to the slow relaxation of some parts of the protein to accommodate the conformational change directed by the reaction coordinate. For example, our unpublished metadynamics simulations of a monomer kinase showed that enhancing the DFG conformational change resulted in distortion of the kinase structure.

## Dimerization restrains and shifts αC inward while increasing the flexibility of DFG

In order to understand why an inhibitor prefers binding with a dimer or monomer BRAF$^{V600E}$, it is important to understand the difference in the conformation and dynamics between the apo monomeric and dimeric BRAF$^{V600E}$. We focus on the αC helix and DFG motif due to their flexibility and importantly specific interactions with the inhibitors (*Figure 2*). Following KLIFS (*Kanev et al., 2021*), the αC position is characterized by the distance between Ile582 on β7 (representing a stable reference point) and the center of mass of the Cα atoms of Asn500, Glu501, and Val502 (representing the center of the αC helix); a distance below 19.6 Å defines the αC-in while a distance above defines the αC-out states. We also examined the salt-bridge formation between the αC-Glu501 and catalytic Lys483; a minimum sidechain distance below 4.5 Å is an alternative way to define the αC-in states (*Tsai et al., 2019*; *Sultan et al., 2018*). These two definitions are consistent and offers complementary information (see later discussion). The holo PHI1-bound structure (PDB: 6P7G) has both protomers resolved with the αC positions of 19.1 and 19.0 Å, suggesting that the αC helix is in but close to the boundary (19.6 Å) with αC-out according to the KLIFS definition (*Kanev et al., 2021*).

Unlike in the co-crystal structures of dimer-compatible inhibitors, the simulations of the apo monomer and dimer revealed that the αC helix mostly samples the αC-out state. Compared to the apo monomer, the αC position is not only more restrained but also shifted inward by about 1 Å in the apo dimer, as seen from the increase of the peak height and the left-shift of the peak position in the probability distributions, from 23.2 to 22.0 Å (*Figure 3a*, *Figure 3—figure supplement 1*). The

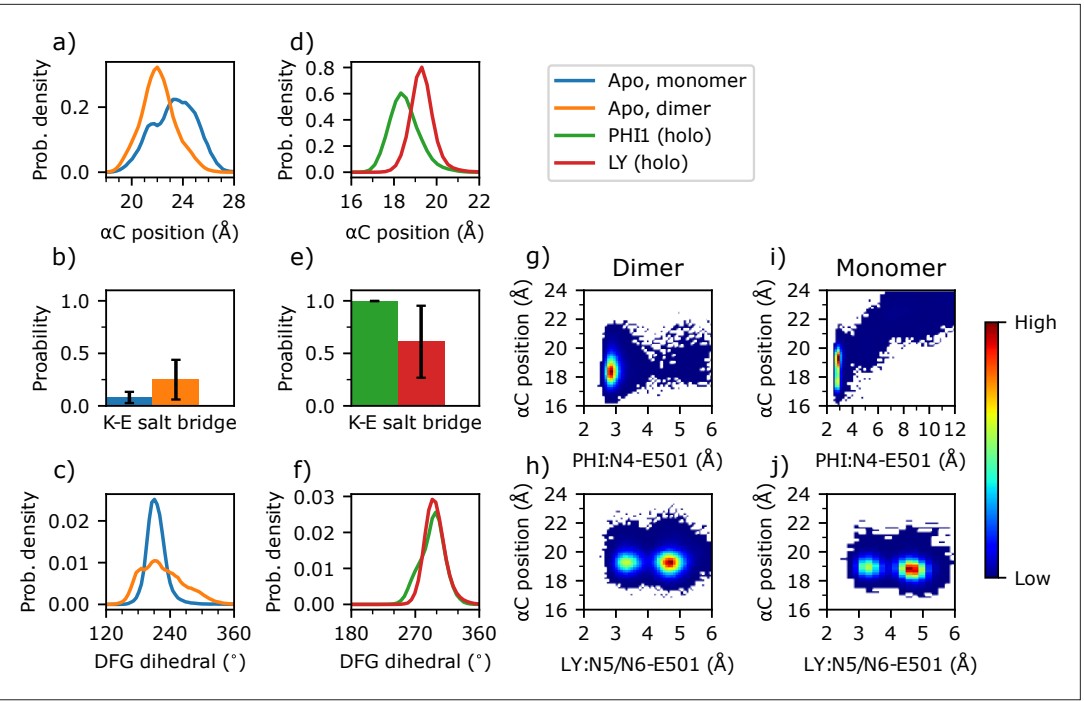

**Figure 3.** Dimerization and inhibitor binding modulate the conformation and dynamics of the αC-helix and DFG motif of BRAF[V600E]. (**a–f**) Probability distribution of the αC position, probability of the Lys483–Glu501 salt bridge, and probability distribution of the DFG pseudo dihedral angle in the apo monomer (blue), apo dimer (orange), PHI1-bound dimer (green), and LY-bound dimer BRAF[V600E] (red). The αC position is defined by the distance between the Cα of Ile582 on β7 and the Cα center of mass of Asn500, Glu501, and Val502 (*Kanev et al., 2021*). A salt bridge between Lys483 and Glu501 is defined by a cutoff distance of 4 Å between the nitrogen of Lys483 and the nearest carboxylate oxygen of Glu501; the standard deviation of the probability across replicas (n=3) are shown as error bars. The DFG pseudo dihedral is defined by the Cα atoms of Ile592, Gly593, Asp594, and Phe595 (*Möbitz, 2015*). (**g–j**) Density plots of the αC position vs. the minimum distance between Glu501 and the amide group of PHI1 (**g ,i**) or LY (**h, j**) in the holo dimer (**g, h**) or holo monomer (**i, j**) BRAF[V600E].

The online version of this article includes the following figure supplement(s) for figure 3:

**Figure supplement 1.** Time series of αC-helix position and DFG dihedral for all apo simulations.

**Figure supplement 2.** Time series of αC-helix position and DFG dihedral for all holo simulations.

**Figure supplement 3.** Time series of E501-ligand hydrogen bonding.

**Figure supplement 4.** Time series of αC-helix position.

**Figure supplement 5.** RMSD time series for the monomer simulations.

flexibility of the αC position in the apo BRAF[V600E] is consistent with a previous MD study (*Maloney et al., 2021*) Enabled by the αC inward movement, the probability of salt-bridge formation between Glu501 and Lys483 is increased by two-fold in the apo dimer (~25%) relative to the apo monomer (~12%, *Figure 3b*, *Supplementary file 1B*). The enhanced but nonetheless unstable Glu501–Lys483 salt bridge indicates that dimerization primes the αC-helix for adopting the αC-in state, for example upon interacting with a dimer-compatible inhibitor.

In contrast to the αC helix, the motion of the DFG motif is significantly enhanced, as evident from the significant widening of the probability distribution of the DFG pseudo dihedral (*Figure 3c*, *Figure 3—figure supplement 1*), defined by the Cα atoms of Ile592 (DFG-2), Gly593 (DFG-2), Asp594 (DFG-Asp), and Phe595 (DFG-Phe; *Möbitz, 2015*). Based on a cutoff of 140°, the DFG pseudo dihedral has been found to discriminate between the DFG-in and DFG-out states of kinases (*Möbitz, 2015*; *Tsai et al., 2019*). Accordingly, the distributions indicate that the DFG motif samples the DFG-out state in both apo monomer and dimer, with the DFG pseudo dihedral of ~210°; however, the dimeric DFG is capable of occasionally sampling the DFG-in state due to the increased flexibility (*Figure 3c*). While this does suggest dimerization loosens the DFG motif, our simulations do not appropriately

model the DFG-out/-in transition as the DFG-in state is only occasionally sampled (*Figure 3—figure supplement 1*).

## PHI1 and LY binding induces the αC-in state to varying degrees and shifts DFG out

Having understood how dimerization modulates the conformational dynamics of the αC helix and DFG motif, we proceeded to explore conformational changes induced by the dimer-compatible inhibitors PHI1 and LY. Interestingly and as expected, both inhibitors further restrain the motion of the αC helix, with its position sampling a narrower range of 4 Å, as compared to 7 Å in the apo dimer (*Figure 3d*, *Figure 3—figure supplement 2*). Importantly, the αC position is shifted inward by at least 2.7 Å in the holo relative to the apo dimer, and PHI1 induces a larger shift, to 18.3 Å as compared to 19.3 Å in the presence of LY (*Figure 3d*, *Figure 3—figure supplement 2*). The inward shift of the αC helix by the two inhibitors is also reflected in the stabilization of the Glu501–Lys483 salt bridge, which is promoted in the presence of LY (60% vs 25% in the apo dimer) and is completely locked in the presence of PHI1 (*Figure 3e*, *Supplementary file 1B*). Although the DFG motif is also significantly restrained through inhibitor binding, the DFG pseudo dihedral in the holo dimer is shifted outward by 80° in complex with either PHI1 or LY (210° in the apo dimer vs. 290° in the holo dimer, *Figure 3f*, *Figure 3—figure supplement 2*).

## H-bond formation with Glu501 is critical for dimer selectivity by shifting αC helix inward

The monomer-selective inhibitors do not contact the center of the αC helix and their co-crystal structures only adopt αC-out state (*Figure 2b*). To test our hypothesis that the h-bond formation with Glu501 is critical for restricting the αC helix to the αC-in states, we examined the density plots of the αC position vs. the distance between the amide nitrogen of PHI1 or LY and the carboxylate of Glu501 in the holo dimer simulations (*Figure 3g, h*, *Figure 3—figure supplement 3*). In the PHI1-bound dimer simulations, the PHI1–Glu501 h-bond is stable with only occasional breakages, as seen from the density maximum centered at the N4–Glu501 distance of 2.9 Å and αC position ~18 Å (*Figure 3g*, *Figure 3—figure supplement 3*).

In the LY-bound dimer simulations, however, the LY–Glu501 h-bond is weaker and less stable than the counterpart of the PHI1-bound dimer, as seen from the local density maximum centered at ~3.4 and the global maximum near ~4.5 Å (*Figure 3g and h*, *Figure 3—figure supplement 3*).

The stronger h-bond between PHI and Glu501 may be attributed to the additional hydrophobic interaction PHI1 forms with Asn500, which is absent for LY (*Figure 2b*). It is also noteworthy that when the PHI1–Glu501 interaction switches from h-bonding to van der Waals interaction, the αC position is slightly shifted outward to ~19 Å, which is similar to the position adopted in the LY-bound dimer simulations. This suggests that the stronger h-bond between PHI1 Glu501 may contribute to the inward αC position as compared to the LY-bound dimer.

To further dissect the mechanism of dimer selectivity, we examined the h-bond interaction between PHI1 or LY and Glu501 in inhibited monomer BRAF$^{V600E}$ simulations. Strikingly, the PHI1–Glu501 interaction can become completely disrupted, with the distance moving beyond 6 Å to as high as 12 Å; correlated with the disruption of the PHI1–Glu501 interaction, the αC position is shifted out to the range of 21 Å–24 Å, similar to that sampled by the apo dimer (*Figure 3i*, *Figure 3—figure supplements 3 and 4*). In stark contrast, the LY–Glu501 interaction remains stable as in the holo dimer simulations (*Figure 3j*, *Figure 3—figure supplements 3 and 4*). These data are consistent with the previous simulations of the LY- and regorafenib (REG)-bound monomeric and dimeric BRAF$^{V600E}$ based on different force fields, which showed that the root-mean-square deviation (RMSD) of the dimer-selective REG is increased in the monomer compared to dimer simulations, whereas the RMSD of the equipotent LY remains the same (*Adamopoulos et al., 2021*).

The correlation between the αC position and the LY–Glu501 interaction confirms our hypothesis that the h-bond interaction between the inhibitor and Glu501 is a key for restraining the αC helix and shifting it to the αC-in states. Since dimerization already restricts the motion of the αC helix and shifts it inward in the apo dimer, inhibitors capable of interacting with Glu501 can bind to the dimer via a conformational selection mechanism in addition to induced fit. On the other hand, conformational selection cannot be exploited for these inhibitors to bind the monomer, as the αC position in

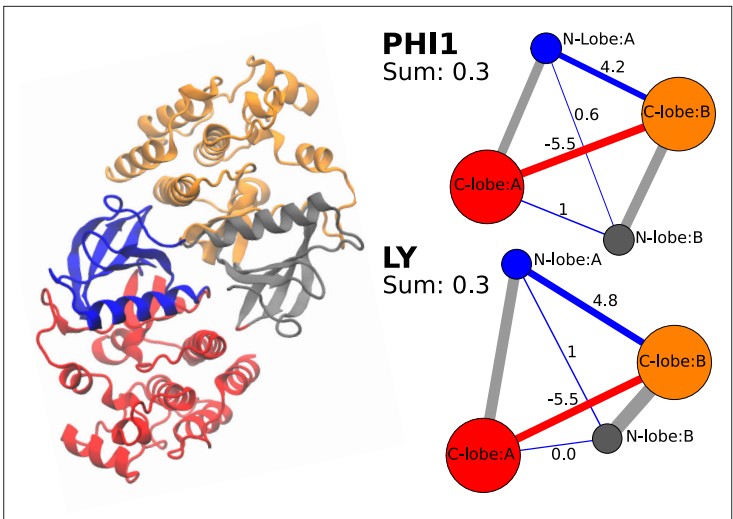

**Figure 4.** Both PHI1 and LY stabilize the interprotomer contacts of BRAF[V600E]. Left. The N-lobe (blue for A; grey for B) and C-lobe (red for A; orange for B) of each protomer in the BRAF[V600E] dimer are separated into different communities according to the difference contact network analysis (*Yao et al., 2018*). Right. The average number of interprotomer contacts was calculated for the apo and holo BRAF[V600E] dimer. (PHI1 top or LY(bottom)). The difference between the holo and apo contacts is shown in the graph form for PHI1 (top) and LY (bottom), and the sum (0.3) is given. Interprotomer contacts are shown as blue (more contacts in holo simulations) or red (more contacts in apo simulations) edges. The difference contact network analysis was performed using the dCNA program (*Yao et al., 2018*). The cutoff distance defining a contact was 4.5 Å; the threshold for determining a stable contact was set to 0.7, and the number of communities was set to 4.

the apo monomer is outward. Compared to the equipotent inhibitors, the dimer-selective inhibitors such as PHI1 form much stronger h-bond with Glu501, which shifts the αC helix further inward. The latter may lead to a larger entropic penalty for the monomer binding as compared to the equipotent inhibitors.

## PHI1 or LY binding has similar stabilizing effect on the dimer interface of BRAF[V600E]

The aforementioned data demonstrates the importance of considering entropic penalty in monomer binding as a contributor to dimer selectivity. To rule out the possibility that the different degree of dimer (de)stabilization may also be a contributing factor for dimer selectivity, we turned to the difference contact network analysis (*Yao et al., 2018*). In this analysis, the BRAF[V600E] dimer was first partitioned into four different communities based on the the residue-residue contacts, which resulted in each community largely corresponding to the N-lobe (blue or grey) and C-lobe (red or orange) of either protomer (*Figure 4* left). Then, a community-community difference contact network between the apo and holo dimer simulation sets was calculated and mapped onto a graph, where the vertices represent the communities and blue and red edges represent the increased and decreased contact probabilities due to inhibitor binding (*Figure 4* right). Since we are interested in testing the dimer stability in the presence of PHI1 or LY, the interprotomer contact probabilities(between N-lobe:A and N-lobe:B or C-lobe:B; between C-lobe:A and C-lobe:B or N-lobe:A) were calculated and summed up. Interestingly, for both PHI1 and LY, the total interprotomer contact probability is increased (by 0.3) in the holo relative to the apo simulations. This net increase is mainly due to the N-lobe:A to C-lobe:B interactions which compensates for the decrease in the C-lobe:A to C-lobe:B contacts. This analysis demonstrates that both the dimer-selective and equipotent inhibitors have the same slightly stabilizing effect on the BRAF[V600E] dimer interface; this rules out the possibility that the dimer selectivity is due to the different degree of dimer stabilization between the dimer-selective and equipotent inhibitors.

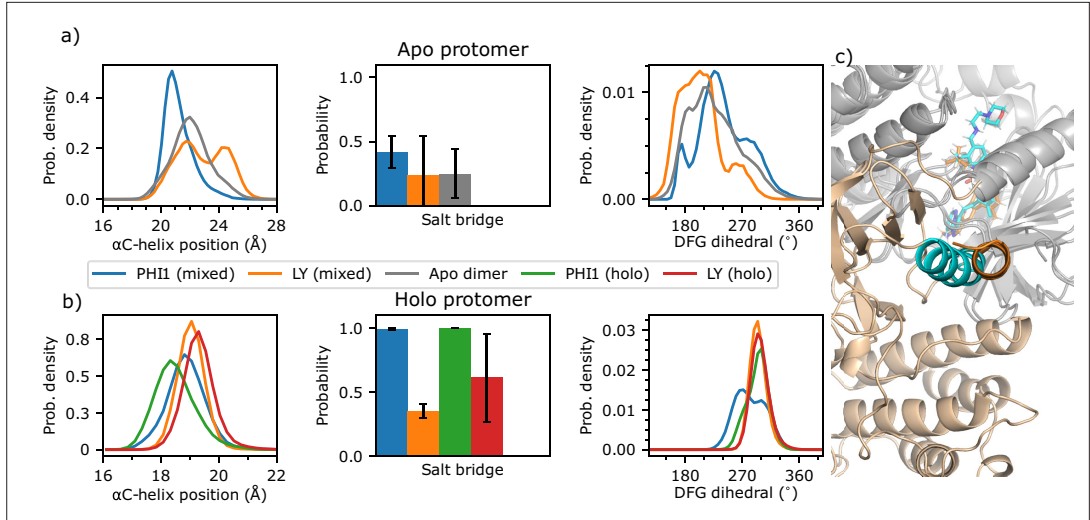

**Figure 5.** Conformation of the αC helix and DFG motif is dependent on the presence or absence of PHI1 in the second protomer. (**a**) The αC helix position, probability of the Glu501–Lys483 salt bridge, and DFG pseudo dihedral of the apo protomer in the one PHI1- (blue) or one LY-bound (orange) mixed dimer simulations. As a reference, the apo dimer data is shown in grey. (**b**) The same quantities as in (**a**) but for the holo protomer in the one PHI1- (blue) or LY-bound (orange) mixed dimer simulations. As a reference, the two PHI1- and LY-bound holo dimer data are shown in green and red, respectively. The standard deviation of the probability across replicas (n=3) is shown in error bars. (**c**) Snapshot from both mixed dimers, after aligning the PHI1- (cyan) and LY-bound (orange) holo protomers (gray). The αC-helix of the apo protomer is highlighted in cyan for PHI1-bound and orange for LY-bound mixed dimer. For simplicity, only the apo protomer from the PHI1-bound mixed dimer is shown.

The online version of this article includes the following figure supplement(s) for figure 5:

**Figure supplement 1.** Time series of αC-helix position and DFG dihedral for all mixed simulations.

**Figure supplement 2.** RMSD time series for the dimer simulations.

## Positive cooperativity of PHI1 is due to the allosteric modulation of the αC and the DFG conformation in the opposite protomer

As previously mentioned, PHI1 was found to exert a more potent inhibition of the second protomers of the BRAF[V600E] dimer, whereas LY demonstrated similar potency in the inhibition of the two protomers (*Cotto-Rios et al., 2020*). To shed light on this cooperativity mechanism, we examined the simulations of the mixed BRAF[V600E] dimers in which only one protomer is in complex with PHI1 or LY. We first compared the αC helix position of the apo protomer in the mixed dimers (*Figure 5a, Figure 5—figure supplement 1*). Surprisingly, the αC helix of the apo protomer in the PHI1-bound mixed dimer is restrained and shifted inward by 1 Å relative to the apo dimer; in contrast, the position of the corresponding αC helix in the LY-bound mixed dimer remains the same but becomes slightly more flexible (blue and grey, *Figure 5a* left, *Figure 5—figure supplement 1*). Consistent with the inward shift of the αC helix, the Glu501–Lys483 salt bridge has a lower average probability and a larger fluctuation in the apo dimer and the apo protomer of the LY-mixed dimer, as compared to the apo protomer of the PHI1-mixed dimer (blue and grey, *Figure 5a* middle).

These data suggest that PHI1 binding in one protomer allosterically modulates the αC helix in the second apo protomer such that it moves inward and becoming more favorable for binding the second PHI1.

While PHI1 or LY binding clearly perturbs the αC helix of the opposite apo protomer, the effect on the DFG conformation is less clear when comparing the DFG dihedral distribution of the the apo protomer in the PHI1 or LY-mixed dimer with that of the apo dimer (blue, orange, and grey, *Figure 5a* right). All three distributions are broad, covering a range of 160–330°. It appears that, relative to the apo dimer, the DFG of the apo protomer in the PHI1-mixed dimer is slightly shifted to the right, whereas that of the LY-mixed dimer is slightly shifted to the left; however, these differences are very subtle and warrant further investigation in future studies.

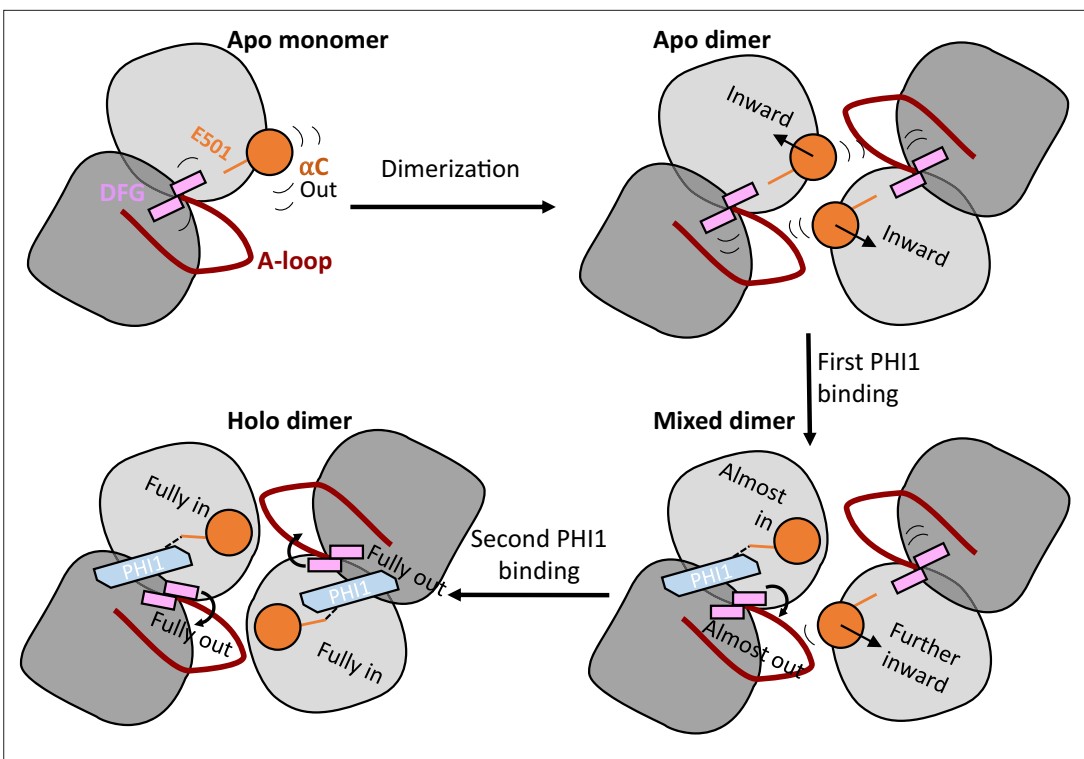

**Figure 6.** A working model that explains dimer selectivity and binding cooperativity of BRAF[V600E] inhibitors. Top left. In the monomeric BRAF[V600E], the αC-helix (orange) is very flexible and exclusively samples the out states. Top right. Upon dimerization, the αC-helix is restrained and shifts inward, while the DFG-motif maintains its conformation but gains significant flexibility. Bottom right. When the first PHI1 molecule binds, its amide linker donates a h-bond to the carboxylate of Glu501 (orange stick) in the first protomer, which locks the αC helix to the αC-in state; it also shifts and restricts the DFG-motif into the DFG-out state through the interaction with the DFG-Asp backbone. The αC-helix in the second unbound protomer is also affected, shifting in the direction of the inhibitor-bound state towards αC-in. Thus, the allosteric pre-organization primes the second protomer for accepting a second PHI1 molecule. Bottom left. When the second PHI1 molecule binds, the αC helix and DFG-motif in both protomers are shifted and fully locked into the αC-in and DFG-out states.

Next, we compared the αC helix position in the holo protomer of the mixed dimers (*Figure 5b* left, *Figure 5—figure supplement 1*). Remarkably, the αC helix in the PHI1-bound protomer of the mixed dimer (blue) is shifted outward by ~1 Å relative to the holo dimer bound to two PHI1 (green, *Figure 5b* left), demonstrating that the second PHI1 binding allosterically shifts the αC helix further inward. Further analysis shows that in the holo dimer the αC helix of one protomer is on average 0.5 Å closer compared to the neighboring protomer (*Supplementary file 1B*). Nonetheless, the Lys483–Glu501 salt bridge is stable in both the mixed and holo dimers; this is because the αC helix predominantly samples the αC-in state in both cases (blue and green in *Figure 5b* middle). In contrast to PHI1, the αC position in the LY-bound protomer of the mixed dimer is similar to that in the LY-bound holo dimer (orange and red in *Figure 5b* left), although the probability of the Lys483–Glu501 salt-bridge in the LY-bound protomer in the mixed dimer is slightly lower than in the holo dimer.

Consistent with the change in the DFG conformation between the holo (two inhibitor) and apo dimers (*Figure 3c and f*), DFG is rigidified upon binding of the first inhibitor, as evident from the narrower DFG dihedral distribution of the PHI1 or LY-bound protomer in the mixed protomer (*Figure 5b* right) compared to the apo protomer in the mixed dimer (*Figure 5a* right). Importantly, the DFG dihedral is right shifted in the occupied vs. apo protomer, demonstrating that the inhibitor pushes the DFG outward.

Consistent with the effect of the second PHI1 on the αC position of the first PHI1-bound protomer, binding of the second PHI1 shifts the peak of the DFG distribution for both protomers further outward, as shown by the 30° larger DFG pseudo dihedral in the holo dimer relative to the mixed dimer (green

and blue in *Figure 5b* right; *Figure 5—figure supplement 1*, *Figure 3—figure supplement 2*). In contrast, there is no significant difference in the DFG pseudo dihedral between the LY-mixed and holo dimers. These data suggest that while the binding of the first PHI1 pushes the DFG outward, binding of the second PHI1 has an allosteric effect, shifting the DFG of the opposite protomer further outward.

## Concluding discussion

We explored the mechanism of dimer selectivity and cooperativity of BRAF$^{V600E}$ inhibitors using MD simulations of the dimeric and monomeric BRAF$^{V600E}$, in the absence and presence of one or two dimer-selective (PHI1) or equipotent (LY) inhibitor(s). The simulations uncovered the atomic details of the remarkable allostery in BRAF$^{V600E}$ dimerization and ligand binding (*Figure 6*), which offer explanation for why some BRAF inhibitors are monomer selective while others are dimer compatible, that is selective or equipotent. Specifically, our data showed that dimerization of BRAF$^{V600E}$ leads to the restriction and an inward shift of the αC helix position relative to the monomer (*Figure 6* top panel), which explains why inhibitors that can stabilize the αC-in states are dimer compatible whereas those that cannot are monomer selective. The fact that both dimerization and inhibitor binding induces αC to move inward contributes to the phenomenon of drug-induced RAF dimerization (*Hatzivassiliou et al., 2010*; *Lavoie et al., 2013*; *Karoulia et al., 2016*).

The co-crystal structure analysis and MD simulations identified a h-bond donor (e.g. an amide linker in the dimer-selective PHI1 or the equipotent LY) as a key for dimer compatibility; the h-bond with the carboxylate of Glu501 stabilizes the αC helix in the αC-in states. Two factors make Glu501 a special and critical anchoring point for inducing the αC-in states. First, it is located at the center of the αC helix, which makes it easier (as opposed to the end of the helix) to induce a helix movement. Second, the h-bonding between the inhibitor and Glu501 is synergistic with the Lys483–Glu501 salt-bridge formation. In contrary, the lack of a h-bond with Glu501, for example in VEM, Debrafenib, or PLX7904, results in the monomer selectivity.

Note, the DFG-in inhibitors can also donate a h-bond to αC-Glu501 and bind the BRAF$^{V600E}$ dimer. An example is SB5909885, which donates a h-bond from the oxime group to αC-Glu501 and also forms a salt bridge with Lys483 (PDB ID: 2FB8; *King et al., 2006*).

The difference between the dimer-selective and equipotent inhibitors is more subtle. The MD simulations revealed that PHI1 forms a more stable h-bond with Glu501 in the BRAF$^{V600E}$ dimer as compared to LY, which is consistent with the ~1 Å inward shift of the αC helix and more stable Lys483–Glu501 salt bridge. The latter differences are much smaller in the co-crystal structures; the αC positions and Lys483–Glu501 distances are only respectively 0.1 and 0.2 Å smaller in the PHI1- vs. LY-bound co-crystal structure. Since the monomeric BRAF$^{V600E}$ has a flexible αC helix that predominantly samples the αC-out states, forming a tighter h-bond would incur a higher entropic penalty for monomer binding. This may explain why the PHI1–Glu501 interaction as well as the αC position are unstable in the monomer simulations but stable in the dimer simulations, in contrast to the LY-bound simulations. Therefore, the stability of the h-bonding with Glu501 may be a key for dimer selectivity.

Without the MD simulations, how would one determine if the h-bond between the inhibitor and Glu501 is stable? We found that the deviation between the αC position and/or K–E distance of the two protomers in the co-crystal structure offers some indication (*Supplementary file 1A*). With the exception of LY and Ponatinib, the αC position and/or K–E distance between the two protomers in the co-crystal structures of AZ628, TAK632, BGB283, SB5909885 deviate by 0.3 Å or higher (*Supplementary file 1A*). In contrast, the αC position and the K–E distance are (nearly) identical between the two protomers in the co-crystal structures of the dimer-selective inhibitors LXH254, RAF709, Sorafenib, and Belvarafenib (*Supplementary file 1A*). The identical αC position and K-E distance in the two protomers suggest that the αC helix is restrained by the inhibitor, that is it forms a stable h-bond with Glu501.

To additionally test this crystal structure-based hypothesis, we examined the co-crystal structures of GDC0879 and Tovorafenib, which were not analyzed in *Adamopoulos et al., 2021*. In the co-crystal structure of GDC0879 (PDB ID: 4MNF), the αC position deviates by 0.3 Å and the K–E distance deviates by 0.1 Å between the two protomers. In the co-crystal structure of Tovorafenib (PDB ID: 6V34), the the αC position deviates by 0.2 Å and the K–E distance deviates by 0.4 Å between the two protomers. These deviations suggest that the αC is not adequately restrained by the inhibitors and therefore we predicted GDC0879 and Tovorafenib to be equipotent. Note, GDC0879 is a DFG-in

inhibitor, which is an additional indication for a equipotent inhibitor. Indeed, both GDC0879 and Tovo-rafenib were found as equipotent in experimental studies (*Karoulia et al., 2016*; *Tkacik et al., 2023*). These analyses led us to propose the following empirical assessment of a RAF inhibitor based on its co-crystal structure with BRAF$^{V600E}$: (1) lack of a h-bond with Glu501 indicates monomer selectivity; (2) presence of a h-bond with Glu501 but inconsistent αC position and/or K–E distance between the two protomers indicates equipotency; (3) presence of a h-bond with Glu501 and identical αC position and K–E distance between the two protomers indicates that the inhibitor is likely (but not necessarily) dimer selective. Given that the resolution of a resolved structure is often ~2–3 Å, this proposed assessment is not intended to replace more rigorous tests, that is utilizing MD simulations.

Finally, the MD analysis uncovered a mechanism for positive cooperativity. Our findings are summarized in *Figure 6*; upon dimerization (top row) the αC-helix goes from αC-out and highly flexible to slightly restrained and inward shifted. The first PHI1 binding in the BRAF$^{V600E}$ dimer restricts the motion of the αC helix and DFG, shifting them slightly inward and outward, respectively (*Figure 6*, bottom right panel). Intriguingly, the first PHI1 binding primes the apo protomer by making the αC more favorable for binding, that is shifting the αC inward (*Figure 6*, bottom right panel). Importantly, upon binding the second PHI1, the αC helix is shifted further inward and the DFG is shifted further outward in both protomers. These data suggest that the positive cooperativity of PHI1 is due to its ability to allosterically modulate the αC and the DFG conformation in the second protomer. Taken together, our findings provide a mechanistic understanding for the remarkable allostery and conformational interplay between kinase dimerization and inhibitor binding. As we prepare the manuscript for submission, a biophysical experiment was published, which suggested that the first inhibitor binding dominates the allosteric coupling between type II inhibitor binding and BRAF dimerization (*Rasmussen et al., 2023*), consistent with our simulation data.

The work presented here has implications for understanding the molecular mechanism of kinase signaling and contributes to the rational design of protomer-selective inhibitors.

## Methods

### Intracellular homogeneous TR-FRET assay

SKMEL239 and SKMEL239-C4 cells were plated at 50,000 cells/well in white TC-treated 96-well plates in 100 µl complete growth media (DMEM). Cells were incubated with the various RAF inhibitors for 1 hr at 37 °C, 5% CO2. ERK phosphorylation was measured using the THUNDER Extreme Phospho-ERK1/2 (T202/Y204) TR-FRET Cell Signaling Assay Kit (Bioauxillium) according to directions for the Standard 2-Plate Assay Protocol for Adherent Cells. Cells were lysed for 30 min at RT under shaking. Lysates were transferred to a white 384-well plate, sealed and incubated with the detection mix antibody at RT for 4 hr. TR-FRET signal was measured at 615 nm and 665 nm excitation using a TECAN SPARK plate reader.

### System preparation for simulations

Simulations were prepared using a crystal structure of BRAF$^{V600E}$ in complex with either PHI1 (PDB ID: 6P7G; *Cotto-Rios et al., 2020*) or LY (PDB ID: 5C9C; *Peng et al., 2015*). The initial structure of the apo simulations was taken from 6P7G. The a-loop is not resolved in either protomer in 5C9C, but is resolved for protomer B in 6P7G. Thus, the missing a-loop in the protomer B of 6P7G and in both protomers in 5C9C were built by rotating and translating the resolved a-loop from the first protomer using the alignment tool in PyMOL (*Schrödinger, 2015*). Chain B (which has the resolved a-loop) was first aligned to chain A using all residues except for the a-loop and the two end residues that connect it to the rest of the protein. Following the alignment, chain B except for the a-loop and its two end residues were deleted. The N-terminus was acetylated and the C-terminus was amidated. Hydrogen atoms were added using the HBUILD facility in the CHARMM package (version c37a2; *Brooks et al., 2009*).

### All-atom continuous constant pH molecular dynamics (CpHMD) simulations

The recently developed all-atom particle mesh Ewald CpHMD (PME-CpHMD) (*Harris et al., 2022*) with the asynchronous pH replica exchange sampling protocol (*Wallace and Shen, 2011*; *Henderson*

*et al., 2020*) was used to determine the protonation and tautomer states of histidines. To prepare for the CpHMD simulations, the histidine residues were first set to HIP with the dummy hydrogens on the Nδ and Nε atoms. The system was solvated in a rectangular water box with at least 10 Å distance between the protein and the boundary (~23,000 water molecules). The protein was represented by the AMBER ff14SB force field (*Maier et al., 2015*) and water by the TIP3P model (*Jorgensen et al., 1983*). The dimer structure was briefly minimized for 500 steps (first 200 were using steepest decent, following 300 used conjugate gradient) with a harmonic force constant of 100 kcal/mol/Å² applied on all heavy atoms of the protein. This was followed by 100 ps of heating to 300 K using the PME-CpHMD simulations at pH 7.0 with the restraints still applied. Once heated the restraints were gradually removed in six stages: in the first two stages the protein heavy atoms were restrained with a force constant of 100 and 10 kcal/mol/Å²; in the next four stages only the backbone heavy atoms were restrained with a force constant of 10, 1.0, 0.1, and 0.0 kcal/mol/Å. Each stage was simulated for 250 ps, for a total of 1.5 ns. A cutoff of 12 Å was used for the nonbonded interactions.

The equilibrated structure was then used to initiate the pH replica exchange PME-CpHMD simulations. The asynchronous pH replica exchange sampling protocol (*Wallace and Shen, 2011*; *Henderson et al., 2020*) was used to accelerate convergence of the coupled protonation and conformational states (*Wallace and Shen, 2011*). Five replicas were created at different pH conditions, from pH 6.5–8.5. Each replica was first equilibrated to its pH by repeating the final four stages of equilibration mentioned above. The pH replica exchange CpHMD was then conducted for 10 ns with attempted swaps of neighboring pH conditions occurring every 2 ps. All other settings are identical to *Harris et al., 2022*. For the calculation of protonation and tautomer state probabilities, the $\lambda$ and $x$ values above 0.8 or below 0.2 were used (default setting in the CpHMD analysis package *Henderson et al., 2022*). At pH 7.5 His477 was protonated at both Nε and Nδ while His510 was protonated at Nε only. These protonation/tautomeric states were used for all convention (fixed-protonation-state) simulations below.

## Conventional fixed-protonation-state MD simulations

Eight BRAF[V600E] systems were simulated, consisting of monomeric and dimeric BRAF[V600E] either in the presence or absence of PHI1 or LY (see *Table 1*). Monomer systems were prepared by eliminating one protomer from the prepared dimer structure. In the apo monomer and dimer systems, ligand(s) was removed. In the mixed or holo systems, one or both inhibitors from the co-crystal structure was kept. The protein was then placed in a rectangular water box with a minimum distance of 10 Å between the protein and edges of the water box using the LEaP program (*Case et al., 2020*). Based on the protonation states determined using CpHMD, sodium and chloride ions were added to neutralize the system and reach a physiological ionic strength of 0.15 M.

The conventional (fixed-protonation-state) MD simulations were carried out using the AMBER20 MD package (*Case et al., 2020*). The proteins was represented by the ff14SB force field (*Maier et al., 2015*) while inhibitors were parameterized by the general AMBER force field (GAFF) method (*Wang et al., 2004*). The TIP3P model (*Jorgensen et al., 1983*) was used to represent water. The Leapfrog integrator was used to propagate the coordinates. The SHAKE algorithm was applied to bonds involving hydrogen to allow for a 2-fs time step. Additionally, the hydrogen mass re-partitioning (*Hopkins et al., 2015*) was used to redistribute the mass between hydrogens and their bonded heavy atoms to allow for a 4-fs time step. A nonbonded cutoff of 8 Å was used as in the ff14SB validation study (*Maier et al., 2015*) while the electrostatic potentials were computed using the particle-mesh Ewald method (*Darden et al., 1993*) with a real-space cut-off of 12 Å and a sixth-order interpolation with approximately 1 Å grid spacing. Each system underwent minimization using 1000 steps of steepest descent followed by 19000 steps of conjugate gradient while the heavy atoms were harmonically restrained using a force constant of 100 kcal/mol/Å². Following minimization, the system was heated to 300 K over 1 ns under an NVT ensemble using a Langevin thermostat (*Feller et al., 1995*) with collision frequency of 1 ps⁻¹ for temperature control. The systems then underwent a 6-stage equilibration in which the backbone restraints were gradually reduced to 10, 5, 2, 1, 0.1 and 0 kcal/mol/Å² over the course of 100 ns under a NPT ensemble. A Monte-Carlo barostat (*Case et al., 2020*) was used to control pressure at 1 bar using a relaxation time of 1.0 ps. Each system were run in three replicates each starting from different random velocity seeds and each run lasted 5 μs.

## Simulation data analysis

CPPTraj (*Roe and Cheatham, 2013*) was used to analyze the protomer conformation (αC-helix position, DFG pseudo dihedral, etc.) and visualizations were produced using PyMOL (*Schrödinger, 2015*). The contact network analysis was conducted using the open source code developed by Yao and Hamelberg (https://github.com/The-Hamelberg-Group/dcna; *Yao et al., 2018*; *The-Hamelberg-Group, 2023*). Unless otherwise noted, the last three μs trajectory frames were used for analysis. All probability distributions were created by combining the last three μs of each replica for each system, with each distribution consisting of 50 bins. Unless specified, distributions contain quantities from both protomers in dimeric simulations.

## Acknowledgements

Funding support by the National Cancer Institute to JS (R01CA256557) and EG (R01CA238229) is acknowledged.

## Additional information

### Funding

| Funder | Grant reference number | Author |
|---|---|---|
| National Institutes of Health | R01CA256557 | Jana Shen |
| National Institutes of Health | R01CA238229 | Evripidis Gavathiotis |

The funders had no role in study design, data collection and interpretation, or the decision to submit the work for publication.

### Author contributions

Joseph Clayton, Data curation, Formal analysis, Investigation, Visualization, Writing - original draft; Aarion Romany, Evangelia Matenoglou, Data curation, Formal analysis; Evripidis Gavathiotis, Formal analysis, Writing – review and editing; Poulikos I Poulikakos, Writing – review and editing; Jana Shen, Conceptualization, Supervision, Funding acquisition, Methodology, Writing – review and editing

### Author ORCIDs

Joseph Clayton http://orcid.org/0000-0002-2652-5994
Evripidis Gavathiotis https://orcid.org/0000-0001-6319-8331
Jana Shen https://orcid.org/0000-0002-3234-0769

Reviewer #1 (Public Review): https://doi.org/10.7554/eLife.95334.4.sa1
Reviewer #2 (Public Review): https://doi.org/10.7554/eLife.95334.4.sa2
Author response https://doi.org/10.7554/eLife.95334.4.sa3

## Additional files

### Supplementary files
MDAR checklist

Supplementary file 1. Supplemental tables. Table A . List of the BRAFV600E inhibitors and the structure features of the co-crystal structures in the PDB. The monomer and dimer selectivities of inhibitors in black are based on the experimental data in *Adamopoulos et al., 2021* and *Cotto-Rios et al., 2020* (PHI1). The monomer and dimer selectivities of inhibitors in red were predicted by us and supported by experiments in *Karoulia et al., 2017* (GDC0879) and *Tkacik et al., 2023* (Tovorafenib). Note, the PDB entries indicated by an astrisk are co-crystal structures in complex with the wild type BRAF (the BRAFV600E forms are unavailable). All structures contain an inhibitor in each protomer, with the exception of PLX7904; in the PDB entry 4XV1, PLX7904

is only present in protomer A. The C helix position (in Å) is defined in the main text. Two values refer to the two protomers. The back pockets (BPs) occupied was calculated by KLIFS (*Kooistra et al., 2016*) based on the definition of *Liao, 2007*. K-E refers to the distance (in Å) between the amine nitrogen of Lys483 and the nearest carboxylate oxygen of Glu501. Table B: Average value and standard deviation of reported quantities from out dimeric BRAF simulations, separated by system and protomer. Each quantity was calculated for each replica after removing the first 2 s for equilibration.

## Data availability

The MD simulation input files and analysis scripts are freely downloadable from https://github.com/JanaShenLab/RAF/ (copy archived at *Clayton and Shen, 2024*). The raw MD trajectories are available on Zenodo (https://doi.org/10.5281/zenodo.14611113).

The following dataset was generated:

| Author(s) | Year | Dataset title | Dataset URL | Database and Identifier |
|---|---|---|---|---|
| Clayton J, Romany A, Shen J | 2025 | AMBER trajectories of dimeric or monomeric BRAF, both in apo and in complex with one of two inhibitors | https://doi.org/10.5281/zenodo.14611113 | Zenodo, 10.5281/zenodo.14611113 |

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
