## [Editor Report · eLife Assessment]

This **fundamental** study illuminates the dynamics of BRAF in its monomeric and dimeric forms, both in the absence and presence of inhibitors, through a **convincing** combination of traditional experiments and sophisticated computational analyses. By revealing novel insights into the selectivity and cooperative processes of BRAF inhibitors, it holds significant promise for the development of future therapeutics, particularly against mutant isoforms in cancer. Overall, these findings will be of great interest to structural biologists, medicinal chemists, and pharmacologists.

---

## [Referee Report · Reviewer #1 (Public Review)]

Summary:

This manuscript from Clayton and co-authors aims to clarify the molecular mechanism of BRAF dimer selectivity. Indeed, first generation BRAF inhibitors, targeting monomeric BRAFV600E, are ineffective in treating resistant dimeric BRAF isoforms. Here, the authors employed molecular dynamics simulations to study the conformational dynamics of monomeric and dimeric BRAF, in the presence and absence of inhibitors. Multi-microseconds MD simulations showed an inward shift of the αC helix in the BRAFV600E mutant dimer. This helped identify a hydrogen bond between the inhibitors and the BRAF residue Glu501 as critical for dimer compatibility. The stability of the aforementioned interaction seems to be important to distinguish between dimer-selective and equipotent inhibitors.

Strengths:

The study is overall valuable and robust. The authors used the recently developed particle mesh Ewald constant pH molecular dynamics, a state-of-the-art method, to investigate the correct histidines protonation considering the dynamics of the protein. Then, multi-microsecond simulations showed differences in the flexibility of the αC helix and DFG motif. The dimerization restricts the αC position in the inward conformation, in agreement with the result that dimer-compatible inhibitors are able to stabilize the αC-in state. Noteworthy, the MD simulations were used to study the interactions between the inhibitors and the protein, suggesting a critical role for a hydrogen bond with Glu501. Finally, simulations of a mixed state of BRAF (one protomer bound to the inhibitor and the other apo) indicate that the ability to stabilize the inward αC state of the apo protomer could be at the basis of the positive cooperativity of PHI1.

---

## [Referee Report · Reviewer #2 (Public Review)]

Summary:

The authors employ molecular dynamics simulations to understand the selectivity of FDA approved inhibitors within dimeric and monomeric BRAF species. Through these comprehensive simulations, they shed light on the selectivity of BRAF inhibitors by delineating the main structural changes occurring during dimerization and inhibitor action. Notably, they identify the two pivotal elements in this process: the movement and conformational changes involving the alpha-C helix and the formation of a hydrogen bond involving the Glu-501 residue. These findings find support in the analyses of various structures crystallized from dimers and co-crystallized monomers in the presence of inhibitors. The elucidation of this mechanism holds significant potential for advancing our understanding of kinase signalling and the development of future BRAF inhibitor drugs.

Strengths:

The authors employ a diverse array of computational techniques to characterize the binding sites and interactions between inhibitors and the active site of BRAF in both dimeric and monomeric forms. They combine traditional and advanced molecular dynamics simulation techniques such as CpHMD (All-atom continuous constant pH molecular dynamics) to provide mechanistic explanations. Additionally, the paper introduces methods for identifying and characterizing the formation of the hydrogen bond involving the Glu501 residue without the need for extensive molecular dynamics simulations. This approach facilitates the rapid identification of future BRAF inhibitor candidates.

---

## [Author Response]

The following is the authors’ response to the previous reviews.

**Reviewer #1 (Public review):**
Comment 1: This manuscript from Clayton and co-authors, entitled ”Mechanism of dimer selectivity and binding cooperativity of BRAF inhibitors”, aims to clarify the molecular mechanism of BRAF dimer selectivity. Indeed, first-generation BRAF inhibitors, targeting monomeric BRAFV600E, are ineffective in treating resistant dimeric BRAF isoforms. Here, the authors employed molecular dynamics simulations to study the conformational dynamics of monomeric and dimeric BRAF, in the presence and absence of inhibitors. Multi-microsecond MD simulations showed an inward shift of the αC helix in the BRAFV600E mutant dimer. This helped in identifying a hydrogen bond between the inhibitors and the BRAF residue Glu501 as critical for dimer compatibility. The stability of the aforementioned interaction seems to be important to distinguish between dimer-selective and equipotent inhibitors.The study is overall valuable and robust. The authors used the recently developed particle mesh Ewald constant pH molecular dynamics, a state-of-the-art method, to investigate the correct histidine protonation considering the dynamics of the protein. Then, multi-microsecond simulations showed differences in the flexibility of the αC helix and DFG motif. The dimerization restricts the αC position in the inward conformation, in agreement with the result that dimer-compatible inhibitors can stabilize the αC-in state. Noteworthy, the MD simulations were used to study the interactions between the inhibitors and the protein, suggesting a critical role for a hydrogen bond with Glu501. Finally, simulations of a mixed state of BRAF (one protomer bound to the inhibitor and the other apo) indicate that the ability to stabilize the inward αC state of the apo protomer could be at the basis of the positive cooperativity of PHI1.

We thank the reviewer for the positive evaluation of our work.

Comment 2a: Regarding the analyses of the mixed state simulations, the DFG dihedral probability densities for the apo protomer (Fig. 5a right) are highly overlapping. It is not convincing that a slight shift can support the conclusion that the binding in one protomer is enough to shift the DFG motif outward allosterically. Moreover, the DFG dihedral time-series for the apo protomer (Supplementary Figure 9) clearly shows that the measured quantities are affected by significant fluctuations and poor consistency between the three replicates. The apo protomer of the mixed state simulations could be affected by the same problem that the authors pointed out in the case of the apo dimer simulations, where the amount of sampling is insufficient to model the DFG-out/-in transition properly.

While the reviewer is correct there are large fluctuations in the DFG pseudo dihedral over the course of the apo simulations, these fluctuations occur primarily in the first 2 *µ*s of the simulations, which were removed from our analysis. The reviewer is also correct that these simulations do not sufficiently model the DFG-out/-in transition; however, a full transition is not necessary for our analysis, as we are only interested in the shift of the DFG pseudo dihedral. As to the reviewer’s comment on the overlapping DFG distributions, we agree that the difference is very subtle. We revised the text.

On page 9, second paragraph from the bottom:

“While PHI1 or LY binding clearly perturbs the *α*C helix of the opposite apo protomer, the effect on the DFG conformation is less clear when comparing the DFG dihedral distribution of the the apo protomer in the PHI1 or LY-mixed dimer with that of the apo dimer (blue, orange, and grey, Figure 5a right). All three distributions are broad, covering a range of 160-330°. It appears that, relative to the apo dimer, the DFG of the apo protomer in the PHI1-mixed dimer is slightly shifted to the right, whereas that of the LY-mixed dimer is slightly shifted to the left; however, these differences are very subtle and warrant further investigation in future studies.”

Comment 2b: There is similar concern with the Lys483-Glu501 salt bridge measured for the apo protomers of the mixed simulations. As it can be observed from the probabilities bar plot (Fig. 5a middle), the standard deviation is too high to support a significant role for this interaction in the allosteric modulation of the apo protomer.

As for the salt bridge, the fluctuation in the apo dimer and LY-mixed dimer is indeed large, and together with the lower average probability suggests that the salt bridge is weaker, which is consistent with the *α*C helix moving outward. To clarify this, we revised the text.

On page 9, second paragraph from the bottom:

“Consistent with the inward shift of the *α*C helix, the Glu501–Lys483 salt bridge has a lower average probability and a larger fluctuation in the apo dimer and the apo protomer of the LY-mixed dimer, as compared to the apo protomer of the PHI1-mixed dimer.”

**Reviewer #2 (Public review):**
Comment 1: The authors employ molecular dynamics simulations to understand the selectivity of FDA approved inhibitors within dimeric and monomeric BRAF species. Through these comprehensive simulations, they shed light on the selectivity of BRAF inhibitors by delineating the main structural changes occurring during dimerization and inhibitor action. Notably, they identify the two pivotal elements in this process: the movement and conformational changes involving the alpha-C helix and the formation of a hydrogen bond involving the Glu-501 residue. These findings find support in the analyses of various structures crystallized from dimers and co-crystallized monomers in the presence of inhibitors. The elucidation of this mechanism holds significant potential for advancing our understanding of kinase signalling and the development of future BRAF inhibitor drugs.The authors employ a diverse array of computational techniques to characterize the binding sites and interactions between inhibitors and the active site of BRAF in both dimeric and monomeric forms. They combine traditional and advanced molecular dynamics simulation techniques such as CpHMD (all-atom continuous constant pH molecular dynamics) to provide mechanistic explanations. Additionally, the paper introduces methods for identifying and characterizing the formation of the hydrogen bond involving the Glu501 residue without the need for extensive molecular dynamics simulations. This approach facilitates the rapid identification of future BRAF inhibitor candidates.

We thank the reviewer for the positive evaluation of our work.

Comment 2: Despite the use of molecular dynamics yields crucial structural insights and outlines a mechanism to elucidate dimer selectivity and cooperativity in these systems, the authors could consider adoption of free energy methods to estimate the values of hydrogen bond energies and hydrophobic interactions, thereby enhancing the depth of their analysis.

As mentioned in our previous response, current free energy methods are capable of giving accurate estimates of the relative binding free energies of similar ligands; however, accurate calculations of the absolute free energies of hydrogen bond and hydrophobic interactions are not feasible yet. Thus, we decided not to pursue the calculations.

**Reviewer #1 (Recommendations to author):**
Comment 1: It would be useful to cite all supplementary figures in the main text (where relevant). In the present version, only Supplementary Figures 2,3, and 4 are cited in the main text.

This was an oversight; supplementary figures 5 through 9 are now cited in the text, to point to the time-series of the quantity discussed. We note that supplementary figures 10 and 11 show the time-series of the root mean squared deviation (RMSD) of each protomer in both all monomeric and dimeric simulations; these quantities are not discussed in the manuscript but are provided for further insight.

Comment 2: It is unclear whether the present data could support a direct involvement of the DFG movement in the allosteric mechanism proposed. The same argument applies to the Lys483Glu501 interaction in the apo protomer of the mixed state simulations. The current simulation data could only support a different stabilization of the αC-helix position. The authors should either remove/tone down the claim or extend the simulations to sample a ”converged” distribution of the DFG dihedral and the Lys483-Glu501 salt bridge of the apo protomers.

We agree that the DFG change in the apo protomer of the PH1-mixed dimer is very subtle (see our response and revision to comment 2); however, the allosteric involvement of DFG is clearly demonstrated in Figure 5 (right panel in 5a and 5b). We compare three states: apo protomer in the mixed dimer, PHI1-bound protomer in the mixed dimer, and holo dimer (i.e., with two PHI1) Binding of the first PHI1 restricts the DFG conformation to the larger DFG dihedrals (blue curves in the top and bottom right panels). This effect (DFG outward and more restricted) is even strong when the second PHI1 binds, locking the DFG in both protomers to a narrow dihedral range 270–330 degree (green and blue curves in Figure 5b, right panel). These are allosteric effects, demonstrating that the second PH1 binding induces conformational change of the DFG in the first protomer. This is why in Figure 6, the DFG of the PHI1-bound protomer in the mixed dimer is labeled as “almost out”, while the DFG in the holo dimer is labeled as “fully out”.

The effect of second PHI1 on the DFG of the first protomer is consistent with that the *α*C helix position, in which case, the second PH1 induces an inward movement of the *α*C of the first protomer (illustrated as “fully in” in the schematic Figure 6). Through the aC movement, the salt-bridge strength is affected, as we discussed in our response and revision to Reviewer’s comment 2a. To clarify these points, we revised the discussion of Figure 5. We made the x axis range of the DFG dihedral distributions the same between the top and bottom panels in Figure 5. To remove the claim of priming effect on DFG, we revised Figure 6.

Page 10, Figure 5:

we made the x axis range of the DFG dihedral distributions on the top and bottom panels the same to facilitate comparison.

Page 11, second and third paragraphs:

“Consistent with the change in the DFG conformation between the holo (two inhibitor) and apo dimers (Figure 3c,3f), DFG is rigidified upon binding of the first inhibitor, as evident from the narrower DFG dihedral distribution of the PHI1 or LY-bound protomer in the mixed protomer (Figure 5b right) compared to the apo protomer in the mixed dimer (Figure 5a right). Importantly, the DFG dihedral is right shifted in the occupied vs. apo protomer, demonstrating that the inhibitor pushes the DFG outward.”

“Consistent with the effect of the second PHI1 on the *α*C position of the first PHI1-bound protomer, binding of the second PHI1 shifts the peak of the DFG distribution for both protomers further outward, as shown by the 30° larger DFG pseudo dihedral in the holo dimer relative to the mixed dimer (green and blue in Figure 5b right; Supplementary Figures 6,9). In contrast, there is no significant difference in the DFG pseudo dihedral between the LY-mixed and holo dimers. These data suggest that while the binding of the first PHI1 pushes the DFG outward, binding of the second PHI1 has an allosteric effect, shifting the DFG of the opposite protomer further outward.”

On page 12, the last paragraph of Conclusion, we remove the claim of the priming effect for DFG:

“The first PHI1 binding in the BRAF^V600E^ dimer restricts the motion of the *α*C helix and DFG, shifting them slightly inward and outward, respectively (Figure 6, bottom right panel). Intriguingly, the first PHI1 binding primes the apo protomer by making the *α*C more favorable for binding, i.e., shifting the *α*C inward (Figure 6, bottom right panel). Importantly, upon binding the second PHI1, the *α*C helix is shifted further inward and the DFG is shifted further outward in both protomers.”

On page 13, Figure 6:

we removed the label “slightly outward” for DFG.

Comment 3: An alternative approach could be using enhanced sampling methods to enhance the diffusion along these coordinates.

We thank the reviewer for bringing up this point. While that the allostery and cooperativity effects are apparent from our simulation data, we agree that enhanced sampling methods in principle could be used to further converge the conformational sampling; however, these approaches face significant challenges. First, BRAF dimer is weakly associated, with *α*C helix forming a part of the dimer interface. Enhanced sampling of *α*C helix would likely result in dimer dissociation. On the other hand, simply using RMSD as a reaction coordinate or progress variable would not necessarily enhance the motion of *α*C helix or DFG or activation loop, which are all coupled. Second, our extensive simulations of a monomer kinase with metadynamics demonstrated that the kinase conformation becomes distorted when a biasing potential is placed to enhance the motion of DFG. This is likely because the other parts of the protein do not have enough time to relax to accommodate the conformational change. To our knowledge, this aspect has not been discussed in the current metadynamics literature, which focuses on the free energy differences and (local) conformational changes along the reaction coordinate. To clarify these points, we added a discussion.

Page 6, end of the first paragraph:

“We note that enhanced sampling methods were not used due to several challenges. First, the BRAF dimer is weakly associated, with *α*C helix forming a part of the dimer interface (Figure 1a). Enhanced sampling (particularly of *α*C helix) would likely lead to dimer dissociation. Second, biased sampling methods such as metadynamics may lead to unrealistic conformational states due to the slow relaxation of some parts of the protein to accommodate the conformational change directed by the reaction coordinate. For example, our unpublished metadynamics simulations of a monomer kinase showed that enhancing the DFG conformational change resulted in distortion of the kinase structure.”

We thank the reviewers again for their valuable comments. We believe our revision has further elevated the quality of the manuscript.